# Factors associated with the lethality of patients hospitalized with severe acute respiratory syndrome due to COVID-19 in Brazil

**Ana Cristina Dias Custódio**[id][º], **Fábio Vieira Ribas**[id][º], **Luana Vieira Toledo**[id][º], **Cristiane Junqueira de Carvalho**[id][‡], **Luciana Moreira Lima**[id][‡], **Brunnella Alcantara Chagas de Freitas**[id]*[º]

Department of Medicine and Nursing, Federal University of Viçosa, Viçosa, Minas Gerais, Brazil

º These authors contributed equally to this work.
‡ CJC and LML also contributed equally to this work.
* brunnella.freitas@ufv.br

**Data Availability Statement:** The data are freely available for public access in the GOV.BR repository. It is maintained by the Brazilian

## Abstract

Due to the high rates of transmission and deaths due to COVID-19, understanding the factors associated with its occurrence, as well as monitoring and implementing control measures should be priority actions in health surveillance, highlighting the use of epidemiological surveillance information systems as an important ally. Thus, the objectives of this study were to calculate the mortality rate of hospitalized patients with severe acute respiratory syndrome due to COVID-19 and to identify factors associated with death, in the period corresponding to epidemiological weeks 01 to 53 of the year 2020. This was a longitudinal study, using the national influenza epidemiological surveillance information system database, routinely collected by healthcare services. The sociodemographic and clinical characteristics of 563,051 hospitalized patients with severe acute respiratory syndrome due to COVID-19 in the five regions of Brazil were analyzed. Cox regression was performed to assess factors associated with patient death during hospitalization. The national lethality rate was 35.7%, and the highest rates of lethality occurred in the Northeast (44.3%) and North (41.2%) regions. During the hospital stay, death was associated with older age (Hazard Ratio—HR = 1.026; p<0.001); male sex (HR = 1.052; p<0.001); living in the North (HR = 1.429; p<0.001), Northeast (HR = 1.271; p<0.001) or Southeast regions of Brazil (HR = 1.040; p<0.001), presenting any risk factor (HR = 1.129; p< 0.001), the use of invasive (HR = 2.865; p<0.001) or noninvasive (HR = 1.401; p<0.001) mechanical ventilation devices. A high case lethality rate was evidenced in patients with severe acute respiratory syndrome due to COVID-19, however, deaths were not evenly distributed across the country's regions, being heavily concentrated in the Northeast and North regions. Older male patients living in the North, Northeast, or Southeast regions of Brazil, who presented any risk factor and were submitted to the use of invasive or noninvasive mechanical ventilation devices, presented a higher risk of evolving to death.

government, with access available to anyone, without any cost and restrictions. The data are publicly available from https://opendatasus.saude.gov.br/dataset/srag-2020.

**Funding:** This study was funded by the "Research Support Foundation of the State of Minas Gerais - FAPEMIG, Edict N° 001/2021 - UNIVERSAL DEMAND, Process: APQ-02360-21".Researcher BACF. The funders had no role in study design, data collection and analysis, decision to publish, or preparation of the manuscript.

**Competing interests:** The authors have declared that no competing interests exist.

## Introduction

The COVID-19 pandemic triggered by a novel coronavirus (SARS-CoV-2) which was detected in the Hubei province, Wuhan city in China, in December 2019, has spread worldwide, requiring from countries and regions all over the world actions to improve health surveillance, especially to stop the spread of the transmitting agent [1]. By April 2020, the confirmed cases of SARS-CoV-2 infection and sickness due to COVID-19 had already exceeded one million people and registered more than fifty thousand deaths worldwide [2].

In Brazil, the Influenza Epidemiological Surveillance Information System, SIVEP-Gripe (*Sistema de Informação de Vigilância Epidemiológica da Gripe*), has been used as the information system for the surveillance of Severe Acute Respiratory Syndrome (SARS). This system has been implemented in response to the Influenza A (H1N1) pandemic since 2009, and it currently also includes the surveillance of hospitalized patients and/or deaths suspected of COVID-19 [3]. SIVEP-Gripe was established as the official channel for notifications of this hazard at the hospital level, reaching a peak of 22,497 hospitalized patients infected with SARS-CoV-2 in the epidemiological week (EW) 20 of 2020—Coronavirus Panel https://covid.saude.gov.br/

The management of the negative impacts caused by this pandemic, especially regarding the reduction and assistance to the high number of victims, has been the most challenging task for healthcare providers and managers. Mortality can be perceived as a multicausal event, influenced by factors inherent to the patients themselves, such as pre-existing clinical conditions, in addition to the structural and organizational issues faced by society and health services [4].

The occurrence of a higher number of deaths due to COVID-19 was verified in the economically disadvantaged population and among those who did not adhere to strategies and measures to control the spread of the virus during the pandemic [5]. Moreover, an association was found between higher mortality rates and the presence of comorbities, such as smoking, diabetes mellitus, hypertension, and obesity [6]. The highest occurrence of deaths due to COVID-19 was associated with the population aged over 60 years in an ecological study involving 113 countries [7]. In Brazil, in the state of Espírito Santo, an association was identified between death and the presence of comorbidities and advanced age among patients receiving medical care in public institutions [8]. In another Brazilian state, Rondônia, the highest occurrence of death was associated with ages over 60 years, male gender, and brown and black skin colors [9].

In this context, due to the high rates of transmission and deaths due to COVID-19, understanding the factors associated with its occurrence, as well as monitoring and implementing control measures should be priority actions in health surveillance, highlighting the use of epidemiological surveillance information systems as an important ally.

Therefore, this study aimed to calculate the case lethality rate of patients hospitalized presenting SARS due to COVID-19 and to identify the risk factors associated with death, in the period corresponding to EW 01 to 53 of 2020.

## Methods

This was an observational, longitudinal study, based on data routinely collected by healthcare services, related to the epidemiological surveillance of patients hospitalized with SARS due to COVID-19 in the period corresponding to EW 01 to 53 of 2020. The study was carried out based on data available in SIVEP-Gripe, which is publicly accessible, non-nominal, without any identification of individuals, available on the following website: https://opendatasus.saude.gov.br/dataset/srag-2020, extracted in the update of April 26th, 2021.

The study population was composed of hospitalized patients presenting SARS due to COVID-19 in 2020, notified in SIVEP-Gripe (n = 691,985). All patients classified with SARS due to COVID-19, who required hospitalization and submitted complete information on the date of outcome (discharge or death) were included. Cases of hospitalized patients with SARS due to other etiologies and notification forms remaining in the system with no outcome of the case were excluded.

In this study, the following sociodemographic, clinical, and diagnostic investigation data available on the individual notification form of hospitalized SARS cases were assessed: age (years), gender (male/female); skin color (white/non-white), level of education (illiterate); elementary school (1st-5th grades); middle school (6th-9th grades); high school (10th-12th grades); higher education; (not applicable); region (North; Northeast; South; Southeast; Midwest); the presence of risk factors (no/yes); description of risk factors (puerperal; chronic cardiovascular disease; chronic hematologic disease; Down's syndrome; chronic liver disease; asthma; diabetes mellitus; chronic neurological disease; chronic pneumopathy; immunodepression; chronic kidney disease; obesity; other morbidities); length of hospitalization (days); admission to Intensive Care Unit (ICU: no/yes); invasive mechanical ventilation (IMV: no/ yes); noninvasive ventilation (NIV: no/yes).

Descriptive and inferential data analysis was performed using IBM SPSS Statistics 23 software, considering a type I error level of 5%. Simple and relative frequencies, measures of central tendency, and dispersion (mean, median, standard deviation, and interquartile values) were presented. The normality of the distribution of the numeric variables was assessed using the Kolmogorov-Smirnov test.

The case lethality rate of SARS due to COVID-19 was calculated based on the proportion of deaths in relation to the total number of patients. The survivor and non-survivor groups were compared for their characteristics. Categorical variables were compared using Pearson's Chi-Squared test. Mann Whitney's test was used to compare numeric variables. Values of $p < 0.05$ were considered statistically significant differences.

For the survival analysis, the dependent variable was the "observation time in days", considering the period between the "patients' admission date", registered in the epidemiological surveillance system, and the "evolution date" of the case, which indicates the date of death (outcome of interest) or discharge, which indicates the end of the observation time (censored).

The Cox Regression analysis, estimating the Hazard Ratio (HR) and its 95% Confidence Interval (CI), was used to evaluate the risk factors for death in patients with SARS due to COVID-19. Univariate and multivariate regression models were used. Variables presenting more than 20% missing information (incompleteness) were not included in the Cox Regression, and only variables presenting more than 80% completeness were included [10].

Since this was a survey that included only public domain data, without participant identification, no approval by the Research Ethics Committee was required.

## Results

During the period evaluated, 563,051 reported cases of patients with SARS due to COVID-19 that required hospital admission had their outcome reported in SIVEP-Gripe. The national case lethality rate was 35.7%. The greatest lethality rates were found in the Northeast (44.3%) and North (41.2%) regions according to the geographical analysis. The Southeast (34.4%), Midwest (31.7%) and South (30.0%) regions presented a lethality rate lower than the nationally calculated value.

Table 1 presents the sociodemographic characteristics of the patients hospitalized with SARS due to COVID, comparing those who were discharged and those who passed away.

**Table 1. Analysis of the sociodemographic characteristics of patients hospitalized with SARS due to COVID-19 in 2020.** Brazil. (n = 563051).

| Variable | General (n = 563051) | Survivors (n = 362144) | Non-survivors (n = 200907) | p-value |
|---|---|---|---|---|
| **Age (n = 563051) – med (Q1-Q3)** | 62 (48–74) | 56 (43–68) | 71 (60–80) | <0.001[a] |
| **Sex (n = 562963) – n (%)** | | | | <0.001[b] |
| Female | 247931 (44.0) | 162399 (44.8) | 85532 (42.6) | |
| Male | 315032 (56.0) | 119682 (55.1) | 115350 (57.4) | |
| **Skin color (n = 442856) – n (%)** | | | | |
| Non-white | 218531 (49.3) | 132670 (47.5) | 85861 (52.6) | <0.001[b] |
| White | 224325 (50.7) | 146913 (52.5) | 77412 (47.4) | |
| **Level of education (n = 208911) – n (%)** | | | | <0.001[b] |
| Illiterate | 14671 (7.0) | 6405 (4.8) | 8266 (10.9) | |
| Elementary school | 56229 (26.9) | 29906 (22.5) | 26323 (34.6) | |
| Middle school | 38172 (18.3) | 23071 (17.4) | 15101 (19.9) | |
| High school | 63531 (30.4) | 45178 (34.0) | 18353 (24.2) | |
| Higher education | 32495 (15.6) | 24928 (18.8) | 7567 (10.0) | |
| Not applicable | 3813 (18) | 3430 (2.6) | 383 (0.5) | |
| **Region (n = 563007) – n (%)** | | | | <0.001[b] |
| North | 45458 (8.1) | 26746 (7.4) | 18712 (9.3) | |
| Northeast | 95460 (17.0) | 53202 (14.7) | 42258 (21.0) | |
| South | 84881 (15.1) | 59431 (16.4) | 25450 (12.7) | |
| Southeast | 284767 (50.6) | 186931 (51.6) | 97836 (48.7) | |
| Midwest | 52441 (9.3) | 35811 (9.9) | 16630 (8.3) | |

Source: SIVEP-Gripe.

Note: med–median; Q1 – 1st quartile (25%); Q3 – 3rd quartile (75%); n – absolute frequency; %—relative frequency; Elementary school - 1st to 5th grades; Middle school - 6th to 9th grades.

[a]Mann-Whitney test

[b]Pearson's Chi-squared test. All calculated values refer to the valid responses total, missing data is not accounted for.

Among the patients who passed away, elderly patients were prevalent, with a median age of 71 years (Q = 60; Q3 = 80), male (57.4%), considered non-white (52.6%), with a low level of education, and who had only attended elementary school (54.5%). Regarding spatial distribution, it is noteworthy that in the Northeast (44.3%) and North (41.2%) regions, the percentage of patients who passed away was higher than in other regions (Southeast 34.4%, Midwest 31.7%, and South 30.0% regions).

From the comparison between the clinical characteristics of surviving and non-surviving patients, it was found that, among the non-survivors, there was a higher proportion of those who presented at least one risk factor (77.3%), remained hospitalized for a longer period, with a median of 10 days (Q1 = 4; Q3 = 18), required ICU admission (63.6%), and required IMV (48.7%), as presented in Table 2.

Table 3 presents the results of the univariate and multivariate Cox Regression, including all variables with completeness greater than 80%. In the univariate analysis, it was observed that the effect of all independent variables was significant to explain the risk of death in patients hospitalized with SARS due to COVID-19. Following multivariate analysis, higher age, increasing the risk with each passing year (HR = 1.026; 95% CI = 1.025–1.026); male sex (HR = 1.052; 95% CI = 1.042–1.062); living in the North (HR = 1.429; 95% CI = 1.397–1.462), Northeast (HR = 1.271; 95% CI = 1.247–1.297) or Southeast regions of Brazil (HR = 1.040; 95% CI = 1.021–1.058), presenting some risk factor (HR = 1.129; 95% CI = 1.115–1.143), using IMV (HR = 2.865; 95% CI = 2.812–2.919) or NIV (HR = 1.401; 95% CI = 1.378–1.425).

**Table 2. Clinical characteristics of patients hospitalized with SARS due to COVID-19 in 2020.** Brazil. (n = 563051).

| Variable | General (n = 563051) | Survivors (n = 362144) | Non-survivors (n = 200907) | p-value |
|---|---|---|---|---|
| **Risk factors/ Comorbidities (n = 563051) – n (%)** | | | | |
| No | 189380 (33.6) | 143814 (39.7) | 45566 (22.7) | <0.001[a] |
| Yes | 373671 (66.4) | 218330 (60.3) | 155341 (77.3) | |
| Puerperium (n = 231611) | 1596 (0.7) | 1322 (0.9) | 274 (0.3) | <0.001[a] |
| Chronic Cardiovascular Disease (n = 306950) | 200840 (65.4) | 112771 (62.9) | 88069 (68.9) | <0.001[a] |
| Chronic Hematologic Disease (n = 232800) | 4477 (1.9) | 2343 (1.7) | 2134 (2.3) | <0.001[a] |
| Down's Syndrome (n = 232517) | 1497 (0.6) | 872 (0.6) | 625 (0.7) | 0.346[a] |
| Chronic Liver Disease (n = 232293) | 5266 (2.3) | 2348 (1.7) | 2918 (3.1) | <0.001[a] |
| Asthma (n = 235855) | 15461 (6.6) | 10986 (7.8) | 4475 (4.7) | <0.001[a] |
| Diabetes mellitus (n = 286688) | 147681 (51.5) | 81704 (48.8) | 65977 (55.2) | <0.001[a] |
| Chronic Neurological Disease (n = 238951) | 23298 (9.8) | 9839 (7.0) | 13459 (13.7) | <0.001[a] |
| Chronic Pneumopathy (n = 238677) | 22845 (9.6) | 10280 (7.3) | 12565 (12.8) | <0.001[a] |
| Immunodepression (n = 234988) | 15386 (6.5) | 7689 (5.5) | 7697 (8.0) | <0.001[a] |
| Chronic Kidney Disease (n = 238497) | 23962 (10.0) | 9912 (7.1) | 14050 (14.3) | <0.001[a] |
| Obesity (n = 237409) | 36254 (15.3) | 22785 (16.1) | 13469 (14.1) | <0.001[a] |
| Other morbidities (n = 279031) | 160081 (57.4) | 90087 (55.2) | 69994 (60.4) | <0.001[a] |
| **Hospitalization Time (n = 563051) med (Q1-Q3)** | 8 (4–14) | 7 (4–12) | 10 (4–18) | <0.001[b] |
| **ICU Admission (n = 511274) – n (%)** | | | | |
| No | 312729 (61.2) | 247299 (74.6) | 65430 (36.4) | <0.001[a] |
| Yes | 198545 (38.8) | 84012 (25.4) | 114533 (63.6) | |
| **IMV (n = 489787) – n (%)** | | | | |
| No | 384436 (78.5) | 296041 (93.2) | 88395 (51.3) | <0.001[a] |
| Yes | 105351 (21.5) | 21530 (6.8) | 83821(48.7) | |
| **NIV (n = 489787) – n (%)** | | | | |
| No | 229166 (46.8) | 127297 (40.1) | 101869 (59.2) | <0.001[a] |
| Yes | 260021 (53.2) | 190274 (59.9) | 70347 (40.8) | |

Source: SIVEP-Gripe.

Note: n – absolute frequency; %—relative frequency; med – mediana; $Q^1$ – 1st quartile (25%); $Q^3$ – 3rd quartile (75%); ICU—Intensive Care Unit; IMV—Invasive Mechanical Ventilation; NIV—Noninvasive ventilation.

[a]Pearson's Chi-squared test

[b]Mann-Whitney test. All calculated values refer to the valid responses total, missing data is not accounted for.

Fig 1 presents the accumulated risk of death of patients hospitalized with SARS due to COVID-19 according to the patients' region of residence and time of hospitalization. Throughout their hospital stay, a higher accumulated risk was observed in the North and Northeast regions, while the South region presented the lowest risk.

## Discussion

This study provided nationwide data on the epidemiology and clinical course during the hospitalization of patients with SARS due to COVID-19. In the epidemiological weeks analyzed, the case lethality rate among patients hospitalized with SARS due to COVID-19 was 35.7%. Patients who were older, male, living in the North, Northeast, or Southeast regions of Brazil, who presented any risk factor and used IMV or NIV were at a higher risk of dying during hospitalization.

**Table 3. Univariate and multivariate Cox Regression analysis of factors associated with the mortality rates of patients hospitalized with SARS due to COVID-19 in 2020.** Brazil. (n = 563051).

| Variable | Univariate Analysis | | | Multivariate Analysis | | |
|---|---|---|---|---|---|---|
| | HR | CI 95% | p-value | HR | CI 95% | p-value |
| **Age (years)** | 1.027 | 1.027–1.027 | <0.001[a] | 1.026 | 1.025–1.026 | <0.001[a] |
| **Sex** | | | | | | |
| Female | 1.000 | - | - | 1.000 | - | - |
| Male | 1.012 | 1.003–1.021 | 0.007[a] | 1.052 | 1.042–1.062 | <0.001[a] |
| **Region** | | | | | | |
| Midwest | 1.000 | - | - | 1.000 | - | - |
| North | 1.341 | 1.313–1.369 | <0.001[a] | 1.429 | 1.397–1.462 | <0.001[a] |
| Northeast | 1.339 | 1.315–1.363 | <0.001[a] | 1.271 | 1.247–1.297 | <0.001[a] |
| South | 0.919 | 0.901–0.937 | <0.001[a] | 0.852 | 0.835–0.870 | <0.001[a] |
| Southeast | 1.075 | 1.057–1.093 | <0.001[a] | 1.040 | 1.021–1.058 | <0.001[a] |
| **Risk Factor Comorbidity** | 1.386 | 1.372–1.401 | <0.001[a] | 1.129 | 1.115–1.143 | <0.001[a] |
| **ICU admission** | 1.567 | 1.552–1.583 | <0.001[a] | 0.950 | 0.938–0.962 | <0.001[a] |
| **IMV** | 2.238 | 2.217–2.260 | <0.001[a] | 2.865 | 2.812–2.919 | <0.001[a] |
| **NIV** | 0.686 | 0.680–0.693 | <0.001[a] | 1.401 | 1.378–1.425 | <0.001[a] |

Source: SIVEP-Gripe.

Note: HR–Hazard Ratio; CI—confidence interval; ICU—Intensive Care Unit; IMV—invasive mechanical ventilation; NIV—noninvasive ventilation.

[a]Significant p<0,05.

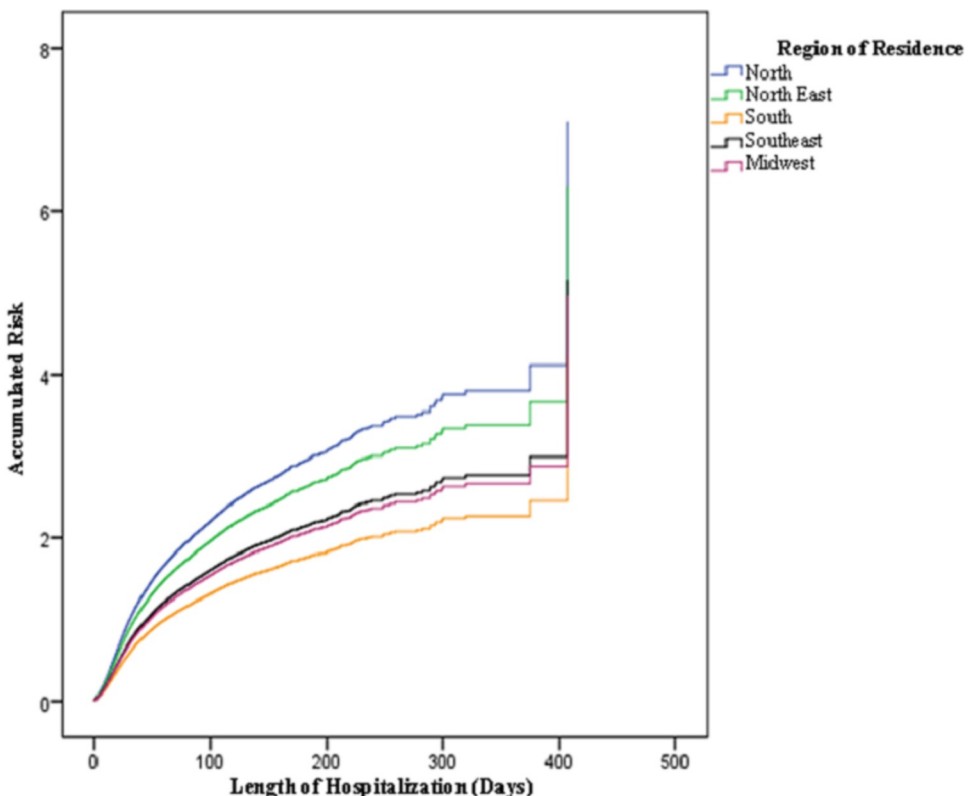

**Fig 1. The accumulated risk of deaths of patients hospitalized with SARS due to COVID-19 in 2020, according to their region of residence and length of hospitalization.**

In another nationwide study, whose data source was the registration of deaths by the Civil Registry Offices, an excess of mortality in Brazil was proven, as early as the onset of the pandemic, in the months of March to May 2020, totaling an excess of 39,146 deaths for the period studied [11]. It should be noted that the analysis of lethality by COVID-19 should consider a combination of factors, such as the intrinsic characteristics of infected individuals (age, previous diseases, lifestyle habits) and the availability of therapeutic resources (hospital beds, healthcare teams, mechanical ventilators, and drugs) [12].

In this study, it has been found that deaths prevailed among the elderly, with a median age of 71 years, and among male patients (57.4%), and that the mortality risk increased with each passing year and for males. A study conducted in the city of Wuhan found that the median age of patients affected by COVID-19 was 60 years, with the greatest severity of disease occurring among those aged 65 years or older. In addition, slightly more than half of the affected patients (50.9%) were male, and these patients developed the most severe form of the disease. Both factors were associated with mortality due to COVID-19 [13]. An Indian study also associated higher mortality rates with ages equal to or greater than 60 years [14]. In another study, also conducted in Wuhan, older age and low lymphocyte count were associated with higher mortality rates among hospitalized patients [15]. The worse outcomes for males were corroborated in a study conducted in Beijing, with a similar prevalence of COVID-19 between men and women, however with a higher risk of death (2.4 times higher) for men, regardless of age [16].

A major research project led by *Global Health 5050* and its partners is working to build the world's largest database on sex and gender and its interface with the health policies in place, especially regarding the COVID-19 pandemic. Among the countries already analyzed, it was observed that there are few or no public health policies related to gender, and those that exist are highly focused on maternal health. Among the confirmed cases of the disease, a higher lethality rate is observed among men when compared to women, even in countries with a higher number of females among the confirmed cases. The higher occurrence of deaths among men may be related to immunological and hormonal issues, such as higher levels of angiotensin-converting enzymes; lower access or use of health services by this population; as well as behavioral issues, such as lower adherence to preventive actions and greater exposure to the virus and other harmful agents such as smoking, which contributes to the development of comorbidities [17]. Sex and gender differences should be accounted for, including in therapeutic interventions for the treatment of COVID-19 [18].

According to this study's results, the fact that these patients presented any comorbidity was associated with higher lethality rates, which is corroborated by several studies, which have identified the association between hypertension, diabetes, chronic obstructive pulmonary disease, heart disease, neoplasms, and HIV and morbidity and mortality due to COVID-19 [19]. The lethality rate is increased by 10.5% for cardiovascular diseases, 7.3% for diabetes, 6.3% for chronic respiratory diseases, and 6% for patients with hypertension [20]. Two studies conducted in Wuhan, an initial focus city for SARS-CoV-2, also identified the presence of comorbidities as one of the factors associated with mortality, with hypertension standing out [13, 21]. A North American study identified that the largest proportion of patients hospitalized due to COVID-19 was composed of males, with advanced age, a history of smoking, and coexisting medical conditions, such as asthma, chronic obstructive pulmonary disease, hypertension, obesity, diabetes mellitus, chronic kidney disease, and cancer [22]. Thus, special attention should be directed to the elderly population, which is also more vulnerable to the development of comorbidities.

Critically ill patients affected by SARS-CoV-2 may present distinct pathophysiological mechanisms, influencing the treatment and the definition of the most adequate ventilatory support strategies for each case [23]. In this study's findings, lethality was associated with the

use of IMV and noninvasive devices. A study conducted in Switzerland, when determining the predictors of hospital mortality related to COVID-19 in elderly patients aged 65 years or older, identified as one of the risk factors the higher requirement of a fraction of inspired oxygen (FiO2) in NIV, that is, each 2% increase in FiO2 added 7% to the risk of death [24]. A study conducted in Italy also identified the high requirement of FiO2 as one of the independent risk factors associated with mortality [25]. However, in Brazil, the negative influences of the absence of a national protocol for the treatment of critically ill patients and the shortage of a properly qualified and trained team for intensive care are highlighted [26].

Differences among the regions in Brazil have been found, highlighting the highest risk of death in the North and Northeast regions, and the lowest risk in the South region. A Brazilian observational, ecological and analytical study, with national coverage, related the mortality of the elderly due to COVID-19 to demographic aspects and income distribution and identified higher mortality in states of the North, Northeast, and Southeast regions. The North region was ranked first place since its large territorial extension and poor transportation routes possibly hinders access to healthcare services [27]. The North region and Northeast region's first-place ranking in mortality can also be explained by the chronic state of social vulnerability in which these populations are found [12]. The scarcity of hospital resources, such as ICU and pulmonary ventilators contribute to the virus lethality rates, a concerning situation which is inherent to the response of healthcare services, with the Northern region having the lowest quantity of these hospital resources [28]. The local, social and demographic characteristics should be accounted for, since Brazil is composed of a large and non-homogeneously distributed population, with cultural and geographical differences, in addition to social inequalities and unequal access to healthcare services [29].

The analysis based on SIVEP-Gripe data enables monitoring the pandemic caused by COVID-19, the definition of strategies for the prevention and control of the disease, and the evaluation of its impact on morbidity and mortality at the national level. Thus, the importance of national health information systems is highlighted as sources of information that support the planning of health policies and programs, contribute to the decision-making process, and allow the evaluation of the impact of interventions. Although these systems present limitations, they are relevant tools for public health, especially for the epidemiological surveillance of diseases [30].

The low completeness in the entries of some variables of SIVEP-Gripe, along with notification errors or delays in feeding the system in some EWs could interfere with the number of cases or deaths. In order to minimize this limitation, the sample included only the notification forms with complete information regarding the outcome date (discharge or death), that is, cases considered closed in the system. Moreover, only variables presenting more than 80% completeness were included in the Cox Regression analysis in order to avoid incorrect inferences. As a strength of this study, the sample size is highlighted, consisting of 563,051 patients hospitalized with SARS due to COVID-19, allowing the tracing of the sociodemographic and clinical epidemiological profile at a national level.

In conclusion, a high case lethality rate was evidenced in patients with SARS due to COVID-19, however, it was found that the fatalities were not equally distributed throughout all regions of Brazil, with the Northeast and North regions presenting the highest lethality rate. Among the factors associated with the occurrence of death, it was found that older patients, who were male, living in the North, Northeast, or Southeast regions of Brazil, who presented any comorbidity and were submitted to IMV or NIV were at a higher risk of death. The recognition of the epidemiological profile from the results obtained can foster decision-making by healthcare providers and managers regarding more effective and equitable interventions, considering the regional diversities found.

## Author Contributions

**Conceptualization:** Ana Cristina Dias Custódio, Fábio Vieira Ribas, Luana Vieira Toledo, Brunnella Alcantara Chagas de Freitas.

**Data curation:** Ana Cristina Dias Custódio, Fábio Vieira Ribas, Luana Vieira Toledo, Cristiane Junqueira de Carvalho, Luciana Moreira Lima, Brunnella Alcantara Chagas de Freitas.

**Formal analysis:** Ana Cristina Dias Custódio, Fábio Vieira Ribas, Luana Vieira Toledo, Brunnella Alcantara Chagas de Freitas.

**Methodology:** Ana Cristina Dias Custódio, Fábio Vieira Ribas, Luana Vieira Toledo, Brunnella Alcantara Chagas de Freitas.

**Project administration:** Ana Cristina Dias Custódio, Fábio Vieira Ribas, Luana Vieira Toledo, Brunnella Alcantara Chagas de Freitas.

**Validation:** Ana Cristina Dias Custódio, Fábio Vieira Ribas, Luana Vieira Toledo, Brunnella Alcantara Chagas de Freitas.

**Visualization:** Ana Cristina Dias Custódio, Fábio Vieira Ribas, Luana Vieira Toledo, Brunnella Alcantara Chagas de Freitas.

**Writing – original draft:** Ana Cristina Dias Custódio, Fábio Vieira Ribas, Luana Vieira Toledo, Brunnella Alcantara Chagas de Freitas.

**Writing – review & editing:** Ana Cristina Dias Custódio, Fábio Vieira Ribas, Luana Vieira Toledo, Cristiane Junqueira de Carvalho, Luciana Moreira Lima, Brunnella Alcantara Chagas de Freitas.

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
