## [Decision Letter · Decision Letter 0]

20 Dec 2021

PGPH-D-21-00716

Factors associated with the lethality of patients hospitalized with severe acute respiratory syndrome due to COVID-19 in Brazil: an ecological study

Dear Dr. Ana Christina

Thank you for submitting your manuscript to PLOS Global Public Health. After careful consideration, we feel that it has merit but does not fully meet PLOS Global Public Health’s publication criteria as it currently stands. Therefore, we invite you to submit a revised version of the manuscript that addresses the points raised during the review process.

We look forward to receiving your revised manuscript.

Kind regards,

Kavitha Saravu, MD, DNB, DTM&H (London)

Academic Editor

Journal Requirements:

1. Please provide separate figure files in .tif or .eps format only.  Please ensure that all files are under our size limit of 20MB.  

For more information about how to convert your figure files please see our guidelines: Once you've converted your files to .tif or .eps, please also make sure that your figures meet our format requirements

2. Please update the completed 'Competing Interests' statement, including any COIs declared by your co-authors. If you have no competing interests to declare, please state "The authors have declared that no competing interests exist"

3. Please note that your Data Availability Statement is currently missing the repository name and/or the DOI/accession number of each dataset OR a direct link to access each database. If your manuscript is accepted for publication, you will be asked to provide these details on a very short timeline. We therefore suggest that you provide this information now, though we will not hold up the peer review process if you are unable.

Additional Editor Comments (if provided):

Dear authors, We request you to revise your manuscript according to reviewers suggestion.

We accept that no consent was required for this publication.

Reviewers' comments:

Reviewer's Responses to Questions

**Comments to the Author**

1. Does this manuscript meet PLOS Global Public Health’s publication criteria? Is the manuscript technically sound, and do the data support the conclusions? The manuscript must describe methodologically and ethically rigorous research with conclusions that are appropriately drawn based on the data presented.

Reviewer #1: Partly

Reviewer #2: Partly

2. Has the statistical analysis been performed appropriately and rigorously?

Reviewer #1: Yes

Reviewer #2: Yes

3. Have the authors made all data underlying the findings in their manuscript fully available (please refer to the Data Availability Statement at the start of the manuscript PDF file)?

Reviewer #1: Yes

Reviewer #2: Yes

4. Is the manuscript presented in an intelligible fashion and written in standard English?

Reviewer #1: Yes

Reviewer #2: Yes

5. Review Comments to the Author

Reviewer #1: The manuscript entitled has novelty and meets publication criteria of PLOS Global Public Health. The manuscript wanted to do address the factors associated with lethality of patients hospitalised with SARS-Cov-2. What could be rationale for conducting ecological study , since the data provided is at individual level . How the study can address the possibility of ecological fallacy.What is the hypothesis generated by the study for which further studies can be employed can be mentioned in recommendations.

Reviewer #2: The abstract needs language correction in Line 24 to 27. The sentence abruptly starts with "To calculate the mortality rate....:. The statistical tests and the methods used are good and appropriate.

However, my concern is about the study design and also about the conclusions drawn. Is the conclusion in line with objectives? Or can it be modified and reported better? What new information or knowledge is this study adding/contributing?

The authors mention that the Ethics approval wasn't required but I would be happier to see that they had approached the committee which then decided to exempt the review and approved it. That makes a stronger case for ethical conduct of research.

6. PLOS authors have the option to publish the peer review history of their article (what does this mean?). If published, this will include your full peer review and any attached files.

**Do you want your identity to be public for this peer review?** For information about this choice, including consent withdrawal, please see our Privacy Policy.

Reviewer #1: No

Reviewer #2: **Yes: **Dr Animesh Jain

---

## [Editor Report · Decision Letter 1]

9 Mar 2022

Factors associated with the lethality of patients hospitalized with severe acute respiratory syndrome due to COVID-19 in Brazil

PGPH-D-21-00716R1

Dear Ana Christina,

We are pleased to inform you that your manuscript 'Factors associated with the lethality of patients hospitalized with severe acute respiratory syndrome due to COVID-19 in Brazil' has been provisionally accepted for publication in PLOS Global Public Health.

Best regards,

Kavitha Saravu, MD, DNB, DTM&H (London)

Academic Editor

Authors have addressed the reviewers comments satisfactorily.